# Resilience Capacity Assessment of the Traditional Lima Bean (*Phaseolus lunatus* L.) Landraces Facing Climate Change

**María Isabel Martínez-Nieto [1], Elena Estrelles [1], Josefa Prieto-Mossi [1], Josep Roselló [2] and Pilar Soriano [1,\***

[1] Jardí Botànic-ICBiBE, Universitat de València, Quart 80, 46008 Valencia, Spain;
maria.isabel.martinez@uv.es (M.I.M.-N.); elena.estrelles@uv.es (E.E.); josefa.prieto-mossi@uv.es (J.P.-M.)

[2] Estación Experimental Agraria de Carcaixent, Carretera CV-5950 (Camino Del Barranquet), Carcaixent, 46740 Valencia, Spain; joseprosello@gmail.com

\* Correspondence: pilar.soriano@uv.es

**Abstract:** Agriculture is highly exposed to climate warming, and promoting traditional cultivars constitutes an adaptive farming mechanism from climate change impacts. This study compared seed traits and adaptability in the germinative process, through temperature and drought response, between a commercial cultivar and Mediterranean *Phaseolus lunatus* L. landraces. Genetic and phylogenetic analyses were conducted to characterize local cultivars. Optimal germination temperature, and water stress tolerance, with increasing polyethylene glycol (PEG) concentrations, were initially evaluated. Base temperature, thermal time, base potential and hydrotime were calculated to compare the thermal and hydric responses and competitiveness among cultivars. Eight molecular markers were analyzed to calculate polymorphism and divergence parameters, of which three, together with South American species accessions, were used to construct a Bayesian phylogeny. No major differences were found in seed traits, rather different bicolored patterns. A preference for high temperatures and fast germination were observed. The 'Pintat' landrace showed marked competitiveness compared to the commercial cultivar when faced with temperature and drought tolerance. No genetic differences were found among the Valencian landraces and the phylogeny confirmed their Andean origin. Promoting landraces for their greater resilience is a tool to help overcome the worldwide challenge deriving from climate change and loss of agrobiodiversity.

**Keywords:** *Phaseolus*; landrace; seed; germination; drought tolerance; genetic approach; sustainable agriculture; climate change

## 1. Introduction

Agriculture is highly exposed to environmental changes, such as climate warming and aridification, as farming activities depend directly on climate conditions. Indeed, the role of agriculture is fundamentally improving natural resources management, rural development, food production and preserving environmental heritage by the conservation of seminatural habitats, landscape and biodiversity [1,2].

Accordingly, the cultivation and conservation of traditional landraces and crop diversification can be effective adaptation strategies to respond to these changing conditions [3], mainly given the increase in aridity and rainfall unpredictability that derive from these changes in environmental conditions.

Loss of crop diversity is a worldwide challenge. Modern cultivars have replaced local landraces, which are now threatened in food production systems, including cultural heritage, local knowledge and

traditional farmer skills. This decline, supported by worldwide globalization, leads to reduced agrobiodiversity on a massive scale, and mainly in developed countries where the industrial food system moves towards genetic uniformity. With the disappearance of traditional species and cultivars, wide ranges of unharvested species also disappear. Promoting local cultivars, which are theoretically more competitive, is one of the major adaptive mechanisms of agriculture to climate change impacts [4–6].

Legumes, specifically *Phaseolus lunatus*, are considered one of the most valuable sources of nutrients in developing countries [7,8]. *Phaseolus lunatus* (Fabaceae), commonly known as "lima bean", and locally termed "garrofó", is the second genus *Phaseolus* species to follow *P. vulgaris* in terms of its economic interest. Attention is paid mainly to its food use worldwide [9], even though other relevant aspects are under study, such as the role on plant protection of the cyanogenic glycosides present in the seeds of this species [10,11].

Cultivated varieties have a South American origin and initially concentrated in northern Peru, where an in-depth selection was developed by the Inca civilization for a long time [12]. According to current germplasm and herbarium records, the conspecific wild ancestor of lima bean is widely distributed from Mexico to Argentina [13]. These landraces are classified into two major groups, Mesoamerican and Andean, according to their geographic origin and seed characteristics [14].

Although it originally comes from Mesoamerica and the Andes, it is currently cultivated throughout Latin America, the southern United States, Canada, and many other world regions including Mediterranean countries, where it is associated with local gastronomy.

On the coasts of the Mediterranean Basin, it is cultivated in warm sunny places in deep well-drained soil. Its strong roots allow plants to thrive on lands where other legumes cannot. It is a highly demanding crop with special requirements. These plants have a type IV climbing growth habit [15] with considerable vegetative development, which means they need a structure that supports, ventilates and illuminates their branches. Nowadays, this cultivation is maintained only for the value of its tender pods and dried grains, and for its special link with traditional cuisine. Currently, the traditional cultivars of this species are being replaced with commercial varieties and represent a testimonial crop in small areas on the European continent.

Hence the present research intends to compare and assess the resistance and adaptability of local cultivars and a commercial variety to face the environmental alterations deriving from climate change. The commercial cultivar is imported from Peru and can be purchased in most retail stores. Primitive landraces, known as 'Pintat', 'Ull de Perdiu' and 'Cella Negra', are traditionally used in the western Mediterranean Basin and are especially cultivated in east Spain (Valencian Community). The use of this species in the eastern Iberian Peninsula in that traditional cuisine is very ancient. Today, we only have references to using these four cultivars in the last 100 years in this region. Our main aim in this work was to recover forgotten crops for the future. In fact, some of the studied cultivars, in particular 'Cella Negra', have practically disappeared today and it has been very difficult to find seeds of this plant.

Furthermore, barcoding is a method to identify taxonomic units using short DNA sequences that allow the determination of the genetic polymorphism and divergences between them. The aim is to identify a region or a combination of regions capable of discriminating taxonomic units, such as species, subspecies, cultivars, or even gene lineages within species [16] and references therein]. Although chloroplast DNA barcoding is utilized mainly to identify plant species, its application can be extended to the food industry, evolution studies and forensics [17]. Various regions of the plastid genome have been proposed to serve as DNA barcodes in plants, such as those put forward by Shaw et al. [18] or Taberlet et al. [19], internal transcribed spacers (ITS) [20] or other specific genes like FRO1 and Phs7 used in legumes phylogenetics by Diniz et al. [21]. This method has been useful in Leguminosae phylogenetics and wild gene pool identifications in *Phaseolus lunatus* [16,21–23]. Thus, it might be a useful tool for typifying local landraces.

This study focuses on seed characterization and providing new information about seed response to temperature and water stress tolerance, estimated during the germinative process, in the *Phaseolus lunatus* traditional cultivars from Mediterranean Europe in line with the future global warming and water deficit scenario.

It also aims to characterize molecularly cultivars—by determining the genetic polymorphism and divergences among local, traditional and commercial, as well as American accessions—of *P. lunatus* in an attempt to genetically delimitate landraces, and to find the potential correlation of these genetic characteristics and germination responses. Moreover, the phylogenetic origin of the Valencian cultivars is studied as part of its molecular characterization.

## 2. Materials and Methods

Four lima bean (*Phaseolus lunatus*) cultivars were tested, three of which were from local Valencian traditional crops ('Pintat', 'Ull de Perdiu' and 'Cella Negra'), mainly provided by the Estación Experimental Agraria de Carcaixent (EEA-Carcaixent) (province of Valencia, Spain). A fourth commercial cultivar imported from Peru to Spain (hereinafter referred to as 'Peru') was bought for the study. The seeds provided by the EEA-Carcaixent were collected during the previous season, nearly one year before starting the germination tests. We did not collect data on the seeds of commercial origin.

### 2.1. Seed Features

Seed dimensions were measured on a digital image using the ImageJ software [24]. Seed weight was determined by an Orion Cahn C-33 microbalance. All the data were obtained from $n = 50$ seeds from each cultivar.

In order to detect differences in variance levels and to identify homogeneous groups, a one-way ANOVA and Tukey's test ($p < 0.05$) were applied, respectively, for each parameter among the different cultivars.

### 2.2. Germination Assays

Seed germination assays were performed with the 'Pintat' and the commercial cultivar, 'Peru', for the low seed availability of the rarest landraces, 'Ull de Perdiu' and 'Cella Negra'. Sporadic tests were conducted with them to provide the initial data for future studies. Data were included as Supplementary data.

Tests were carried out using four replications of 10–15 seeds (depending on seed availability) per treatment for each cultivar. Tests were conducted on 14-cm diameter Petri dishes with paper filters kept in climate-controlled cabinets. Illumination was provided by daylight fluorescent tubes with a 12-h photoperiod and a mean irradiance of 100 $\mu mol \cdot m^{-2} \cdot s^{-1}$. The germination process was evaluated for 15 days. Germinated seeds were counted daily.

Firstly, the optimum germination conditions for successive experiments were set. Temperature screening, using six constant temperatures (15 °C, 20 °C, 25 °C, 30 °C, 35 °C, 40 °C), was applied to determine the optimal germination temperature.

The water stress effect was evaluated by the controlled osmotic potential levels generated using polyethylene glycol (PEG 6000) solution at 30 °C according to Villela et al. [25] to obtain 0 (control), −1, −2, −3, −4 and −5 bar. In order to minimize the evaporation and concentration of the effect of solutions and to maintain the known osmotic potential stable, seeds were moistened every 24 h with fresh PEG solutions and plates were kept in double plastic zip lock bags. After 15 days, non-germinated seeds were transferred to distilled water. Thereby, germination capacity recovery was tested to check the potential influence of PEG exposure on seed germination behavior of *Phaseolus lunatus* cultivars.

Germination Percentage and Mean Germination Time (MGT) were considered to compare seed responses. The base temperature (Tb), by back extrapolation [26], and the thermal time requirement [27] were also calculated to compare thermal responses. Then, the base potential ($\Psi b$) and hydrotime ($\theta$) for each cultivar were calculated [28,29].

Variance levels and homogeneous groups were determined by the one-way ANOVA and Tukey's test ($p < 0.05$), respectively, for each parameter among cultivars.

*2.3. Genetic Assays*

2.3.1. Plant Material and DNA Extraction

The plant material used in the molecular analysis was obtained from the seeds germinated in Germplasm Bank (UV) or Estación Experimental Agraria (EEA-Carcaixent). Eight individuals of each cultivar were analyzed, except for 'Cella Negra', where only five individuals were available at the time of when the genetic assays were done. For the 'Ull de Perdiu' and 'Pintat', we studied two different samples; one seed accession was obtained from the EEA-Carcaixent, while the other was bought from a traditional market. All the 'Cella Negra' seeds came from EEA-Carcaixent and all the 'Peru' ones were obtained from a market as local farmers do not traditionally cultivate them. All the accessions were identified according to seeds' distinctive morphological features. DNA was extracted from young leaves using Doyle and Doyle [30] protocol, modified by Soltis laboratory (2002; https://www.floridamuseum.ufl.edu/wp-content/uploads/sites/95/2014/02/CTAB-DNA-Extraction.pdf). In order to phylogenetically locate the Valencian cultivars, all the accessions provided by Serrano-Serrano et al. [22] in NCBI were used.

2.3.2. Molecular Analyses

A pool of five chloroplast and three nuclear markers (see Table 1) was analyzed to characterize the Valencian landraces. These markers were variable in other studies related specifically to *Phaseolus lunatus* or *Phaseolus* spp. [21–23]. A standard PCR protocol following GoTaq® Polymerase (Promega, Madison, WI, USA) instructions was used for all the markers, except for *Phs7* and *FRO3*, which were amplified following Diniz et al. [21]. The PCR products were purified using the Real Clean PCR Kit (Durviz, Valencia, Spain) and sequenced in an ABI 3100 Genetic Analyzer with the ABI BigDye Terminator Cycle Sequencing Ready Reaction Kit (Applied Biosystems, Foster City, CA, USA).

**Table 1.** The markers analyzed for the *P. lunatus* Valencian cultivars, primer names, Tm (primer melting temperature) and original references in which they were described.

| Marker | Primer Names | Tm | Reference |
|:---:|:---:|:---:|:---:|
| *atpB-rbcL* | atpB-f / rbcL-r | 55 | [31] |
| *trnL-trnF* | trnL (UAA) 3′ exon f / trnF (GAA) r | 58 | [19] |
| *trnL* intron + *trnL-trnF* | trnL (UAA) 5′ exon f / trnF (GAA) r | 58 | [19] |
| *rpoB-trnC* | rpoB-f / trnC-r | 55 | [18] |
| *psbA-trnH* | psba-f / trnH-r | 56 | [32] |
| ITS | ITS1-f / ITS4-r | 58 | [20] |
| *Phs7* | Phs7-f / Phs7-r | 62 | [21] |
| *FRO3* | FRO3-f / FRO3-r | 62 | [21] |

2.3.3. Phylogenetic Analyses

As Serrano-Serrano et al. [22] used ITS, *Atpb-rbcL* and *trnL-trnF* fragments to construct a wide *P. lunatus* phylogeny, these fragments were employed to locate the origin of the Valencian cultivars. Two individuals of each Valencian landrace and all the accessions provided by Serrano-Serrano et al. [22] in NCBI were analyzed. MAFFT v. 7.402 [33,34] was utilized to generate a multiple sequence alignment. The preconfigured MAFFT strategy, which favors accuracy with the FFT-NS-I algorithm (an iterative refinement method that performs 1000 iterations), and default parameters were selected. The ambiguously aligned regions were automatically dealt with using GBlocks v. 091b [35] by implementing the least stringent parameters, but allowing for gaps in 50% of sequences. (NCBI accession numbers: MT072230–MT072258, ITS; MT080626–MT080654, *atpB-rbcL*; MT090972–MT091000, *trnL-trnF*; MT110491–MT110519, *rpoB-trnC*; MT124955–MT124983, *psbA-trnH*; MT154089– MT154117, phs7; MT154118–MT154146, FRO3).

A Bayesian phylogenetic MCMC analysis was run with MrBayes v. 3.2.2 [36]. Indels were coded with SeqState v. 1.4.1 [37] according to modified complex coding. The coded indels were considered to be a partition of standard data (states = 0, 1, 2, 3, ?), with the gamma rate and hyperprior fixed at 1.0 to allow different stationary state frequency proportions to be explored by the MCMC procedure. The optimal substitution models for the nucleotide section were inferred with PartitionFinder2 [38] by considering a model with linked branch lengths for the codificant and non-codificant regions of nrITS and chloroplast fragments, respectively, and using the Bayesian information criterion (BIC). Finally, three partitions were considered: two within the ITS (ITS1 + ITS2 and 5.8S), as well as *Atpb-rbcL + trnL-trnF*. This analysis favored the HKY + G model for the ITS1 + ITS2 partition, K80 + I for 5.8S, and also GTR + I for the chloroplast region. Then, a MrBayes analysis was conducted with two parallel and simultaneous four-chain runs, executed over $5 \times 10^6$ generations, starting with a random tree, and sampling after every 500th step. The first 25% of the data was discarded as burn-in. The 50% majority-rule consensus tree and the corresponding posterior probabilities were calculated from the remaining trees. Chain convergence was assessed by ensuring that the average standard deviation or split frequencies (ASDSF) values were below 0.01, and the potential scale reduction factor (PSRF) values approached 1.00. iTOL v. 4.4.2 [39,40] was used to construct the 50% majority rule consensus tree. The programs MAFFT, MrBayes and PartitionFinder2 were hosted at the CIPRES Science Gateway [41].

2.3.4. DNA Polymorphism and Divergence

The MAFFT original alignment without outgroups was employed to evaluate DNA polymorphism and divergence by taking in account the studied Valencian cultivars and all the accessions, including those used by Serrano-Serrano et al. [22], respectively. All the analyzed markers were utilized to study the Valencian landraces, as well as ITS, *Atpb-rbcL* and *trnL-trnF*, for the whole analysis. Five parameters were calculated by DnaSP v. 6 [42]: segregating sites (s), nucleotide diversity ($\pi$), number of haplotypes (h), haplotype diversity ($H_d$), and the nucleotide genetic differentiation estimate $K_{st}$.

## 3. Results

### 3.1. Seed Features

The seed dimensions of these four *Phaseolus lunatus* cultivars were similar (Table 2). It is noteworthy that the 'Pintat' seeds obtained higher values for the length and width parameters, and had a more rounded contour. The thickness analysis indicated significant differences among cultivars, with the lowest values for the traditional landraces. The 'Peru' and 'Pintat' seeds were the heaviest, while the 'Cella Negra' seeds were lightweight.

Seed coat color is an important consumer trait. In this group, it is a relevant distinctive character for these traditional cultivars (Figure 1; Table 2). The studied commercial cultivar, identified herein as 'Peru', has a completely white seed coat showing no type of pigmentation. The traditional 'Pintat'

depicts an irregular spotted pigmentation over the whole external cover, from dark maroon to brown, depending on the maturation stage. The 'Ull de Perdiu' cultivar has a characteristic black eye surrounding the hilum seed zone. Finally, the cultivar known as locally 'Cella Negra' is identified by having a dark brown to black seed tip close to the embryo radicle lobe.

**Table 2.** Seed morphological features for the different studied cultivars. Length (L), width (W) and their relation (L/W), thickness, as well as weight and color trait of seed coat, are indicated. The same letters indicate homogeneous groups among temperatures ($p < 0.05$) for each cultivar.

|  | 'Peru' | 'Pintat' | 'Ull de Perdiu' | 'Cella Negra' |
|---|---|---|---|---|
| **L (mm)** | 25.3 ± 0.21 b | 26.4 ± 0.13 a | 24.8 ± 0.17 b | 25.0 ± 0.18 b |
| **W (mm)** | 15.5 ± 0.14 b | 17.3 ± 0.09 a | 15.7 ± 0.12 b | 15.9 ± 0.14 b |
| **L/W** | 1.63 ± 0.16 a | 1.53 ± 0.10 b | 1.58 ± 0.14 ab | 1.58 ± 0.21 ab |
| **Thickness (mm)** | 6.72 ± 0.45 a | 5.60 ± 0.42 c | 5.96 ± 0.81 bc | 6.37 ± 0.72 ab |
| **Weight (g)** | 1.82 ± 0.07 a | 1.81 ± 0.10 a | 1.75 ± 0.39 ab | 1.54 ± 0.15 b |
| **Pigmentation** | No | Yes | Yes | Yes |

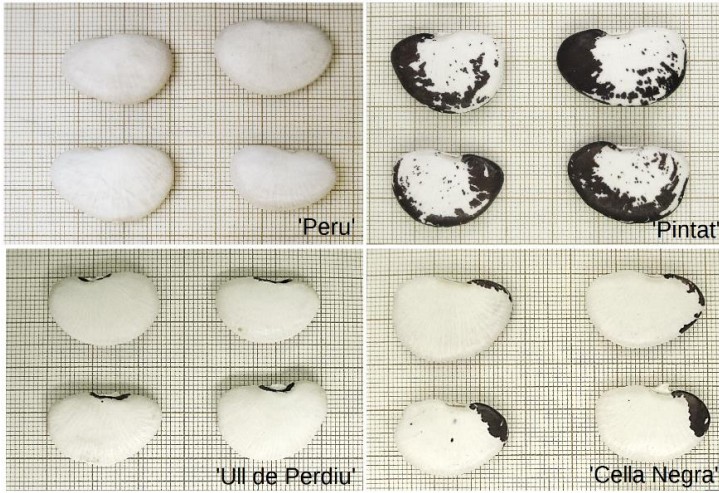

**Figure 1.** Seed morphological traits and pigmentation for the studied *Phaseolus lunatus* cultivars; 'Peru'; 'Pintat'; 'Ull de Perdiu'; 'Cella Negra'.

*3.2. Germination Assays*

3.2.1. Germination Response to Temperature

High germination percentages were achieved at almost all the tested temperatures. The lowest values were for 35 °C in the two studied cultivars, while no germination was observed in any of them above this temperature.

After taking into account the values obtained for the germination percentage and mean germination time, the optimal germination temperature for the studied group of *Phaseolus lunatus* cultivars was set at 30 °C (Figure 2; Table 3). Good results for germination percentages were also obtained at 15 °C and 25 °C, mainly for the 'Pintat' cultivar, but germination was slower in both cases. The values with the same letters did not significantly differ at the 5% level. No significant differences were found when comparing germination velocities among the cultivars at each specific temperature.

The regression lines, indicating the response of germination velocity to increasing temperature (Figure 3), showed a steeper slope for the local 'Pintat' cultivar than for the commercial one, labelled as 'Peru', given the shorter mean germination time; i.e., faster germination. This effect became evident at the temperatures exceeding 19 °C. When the thermal time, S and Tb parameters were calculated from the regression line equations, the 'Pintat' seeds gave values of 131.6 °C·day$^{-1}$ and 5.2 °C respectively, with 185 °C·day$^{-1}$ and −15.0 °C for 'Peru'.

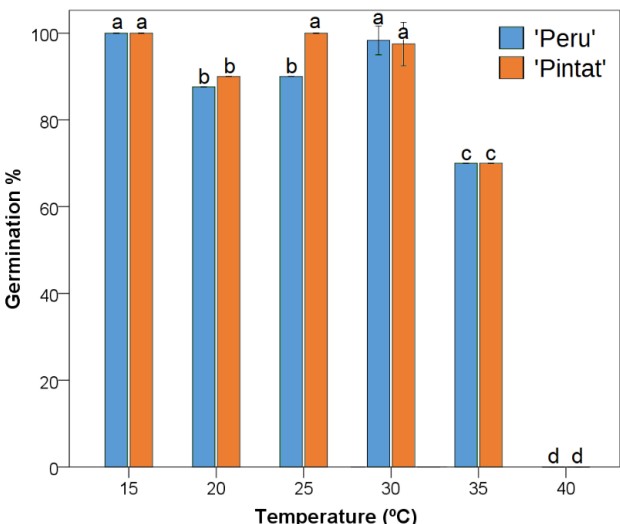

**Figure 2.** The germination percentage values obtained at different temperatures for the studied *Phaseolus lunatus* cultivars. The same letters indicate homogeneous groups ($p < 0.05$).

**Table 3.** Mean germination time (days) at different temperatures (°C) for 'Pintat' and 'Peru' cultivars. The same letters indicate homogeneous groups among temperatures ($p < 0.05$) for each cultivar.

|  | **15 °C** | **20 °C** | **25 °C** | **30 °C** | **35 °C** | **40 °C** |
|---|---|---|---|---|---|---|
| 'Peru' | 5.6 ± 0.2 b | 5.9 ± 0.5 b | 5.0 ± 0.3 b | 3.9 ± 0.6 a | 5.1 ± 0.6 b | - |
| 'Pintat' | 6.1 ± 0.2 cd | 5.6 ± 0.2 c | 4.6 ± 0.2 b | 3.6 ± 0.4 a | 6.3 ± 0.5 d | - |

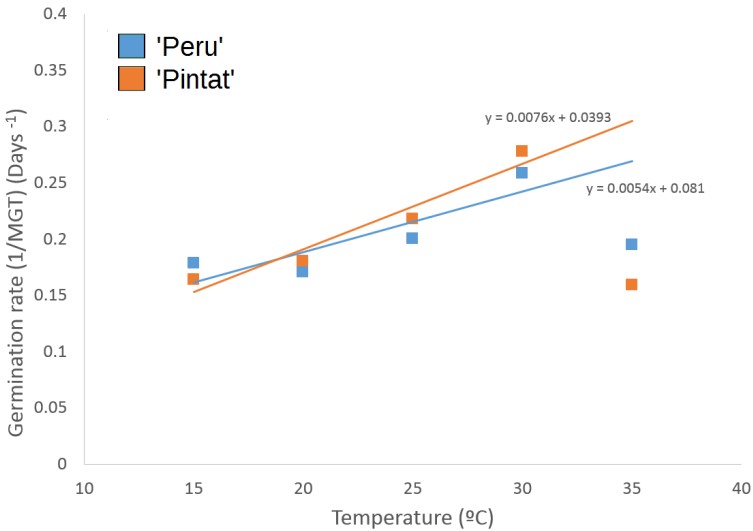

**Figure 3.** The linear regression of the germination rates (MGT) related to the tested temperatures for two cultivars: 'Pintat' and 'Peru'.

### 3.2.2. Germination Response to Drought Stress

Characteristically, germination was affected by rising PEG concentrations. In both cases, a drastic reduction in germination was recorded from −4 bars, and no germination took place at −5 bar. However, Figure 4 and Table 4 show better tolerance to induced water stress for the 'Pintat' cultivar, which obtained higher germination percentages and velocity under all the tested conditions. At −2 bar, no significant differences appeared in relation to the control for 'Pintat' landrace, while germination lowered by 28.8% for the cultivar 'Peru'.

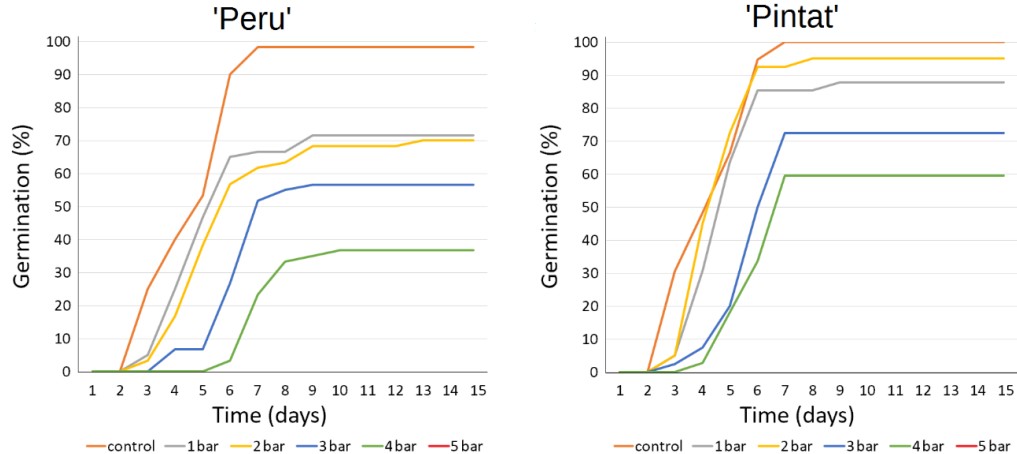

**Figure 4.** The accumulative germination percentages of the studied cultivars at increasing osmotic pressures obtained with PEG from 0, the control, to a maximum of −5 bar.

**Table 4.** Mean germination time (MGT), expressed as days, for the *Phaseolus lunatus* cultivars analyzed at increasing PEG 6000 concentrations. The same letters indicate homogeneous groups among cultivars and the tested concentrations ($p < 0.05$).

| | Osmotic Potential (Bar) | | | | |
|---|---|---|---|---|---|
| | **0** | **−1** | **−2** | **−3** | **−4** |
| 'Peru' | 3.9 ± 0.6 a | 4.2 ± 0.4 ab | 4.8 ± 1.2 ab | 5.5 ± 0.6 abc | 6.5 ± 0.6 bc |
| 'Pintat' | 3.6 ± 0.4 a | 3.9 ± 0.8 a | 3.8 ± 0.2 a | 5.1 ± 0.2 ab | 5.2 ± 0.6 ab |

The germination test conducted at increasing water stress pointed out differences in the seeds of the studied cultivars for their physiological potential to face water deficit. A drastic drop in germination was recorded at −4 and −5 bars. The cultivar 'Pintat' demonstrated better tolerance to water stress, which obtained values above 50% for the germination percentage for all the tested osmotic potentials up to −4 bar.

The 'Pintat' cultivar displayed a faster response to germination velocity under all the conditions, and only showed a clear decrease from −2 bar (Table 4; Figure 5).

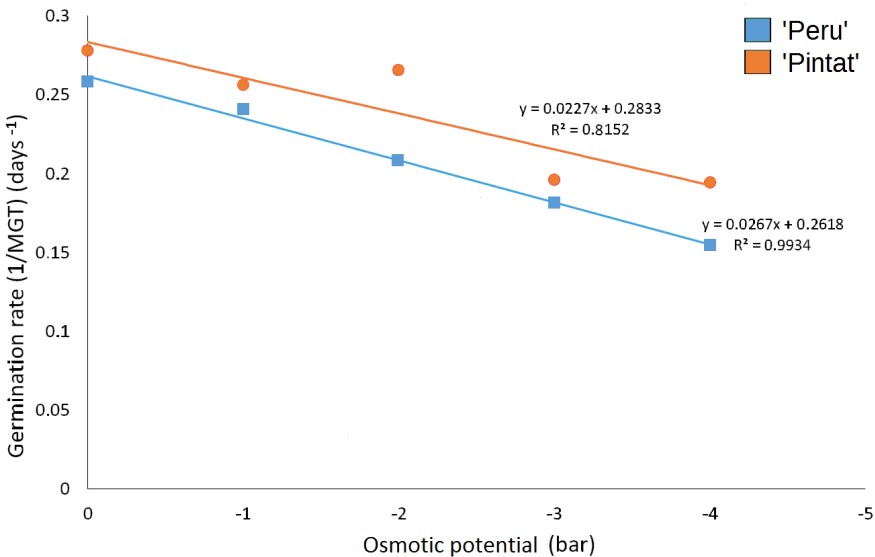

**Figure 5.** The relation between osmotic potential (bar) and germination rate (1/MGT) for the studied cultivars at 30 °C.

The hydrotime calculated from the linear regression slope was 37.5 and 44.1 bar·day for the cultivars 'Peru' and 'Pintat', respectively. The theoretical values calculated for the minimum osmotic potential ($\Psi$b) at which radicle emergence was prevented were respectively −9.8 and −12.5 bar for these same cultivars. When PEG exposure ended, non-germinated seeds were transferred to the non-stressed medium. After 15 days of incubation in distilled water, no recovery was observed at any tested concentration.

### 3.3. Genetic Assays

The dataset herein considered comprised new 29 sequences, including three nuclear and five chloroplastic concatenated fragments that belong to the four more common *P. lunatus* cultivars in Spain. The phylogenetic analyses included the ITS, *Atpb-rbcL* and *trnL-trnF* fragments, two individuals of each Valencian landrace and all the accessions provided by Serrano-Serrano et al. [22] in NCBI. Seventy-eight individuals were analyzed. The MAFFT algorithm produced an alignment of 1828 bp with outgroups and 1410 without them. After the automatic removal of ambiguously aligned positions in GBlocks v. 0.91b, 97% (1781 nucleotides) of the original length, 13 selected blocks were kept after taking the outgroups into account. This final alignment included 110 variable positions, of which 73 were parsimony informative and 37 were singletons. The MrBayes analysis reached an average standard deviation of split frequencies of 0.01 after 156 generations. The resulting topology is presented in Figure 6, where the Valencian cultivars were clustered in the AI gene pool, together with the Andean Cordillera accessions from Ecuador and Peru with high clade support (BI ≥ 0.9). These landraces also formed a high supported clade inside the AI gene pool (BI = 0.98). The main groups also displayed good clade support (BI ≥ 0.9), except for the MII gene pool (BI = 0.61), which was clustered in a wider and well-supported Mesoamerican group, split inside.

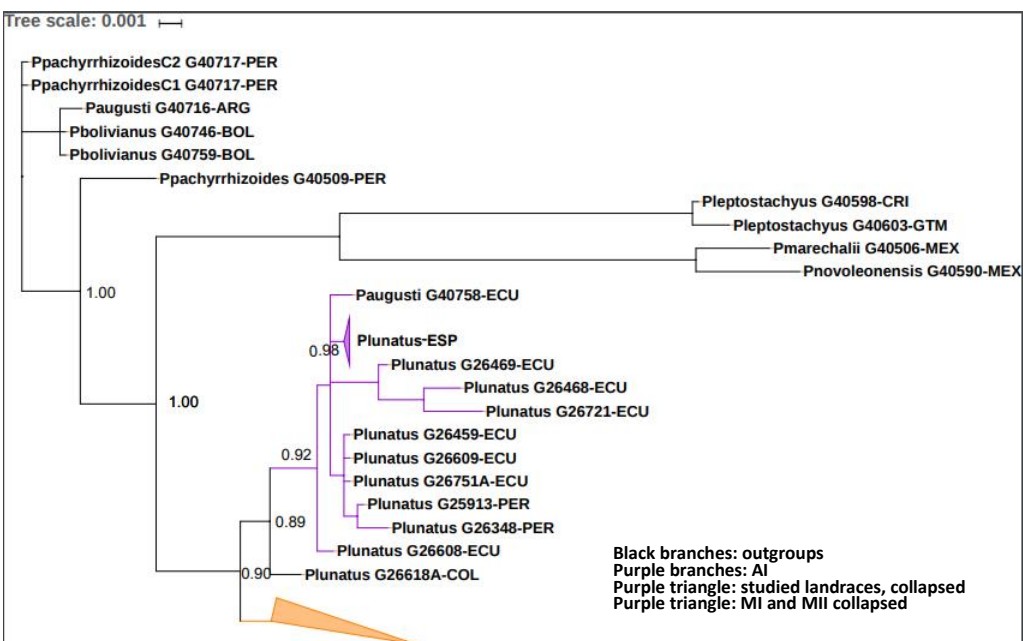

**Figure 6.** Phylogram depicting the phylogenetic relations among the *P. lunatus* accessions from Spain and South America obtained with MrBayes and based on nrITS and cpDNA data. Support values are given for the main nodes (BI). Colors correspond to the gene pools for wild *P. lunatus*: black branches belong to outgroups, purple branches to AI (Andean I) and the purple triangle inside represents collapsed clades of Valencian landraces (local and 'Peru') as they were almost genetically identical, the orange triangle represents the MI (Mesoamerican I) and MII (Mesoamerican II) collapsed clades as they did not provide any relevant information for our purposes. The whole tree is shown in the Supplementary Material (Figure S2). COL = Colombia, ECU = Ecuador, PER = Peru, ESP = Spain.

Polymorphism and divergence analyses were conducted throughout two groups: only the Valencian cultivars and Valencian and South American cultivars from Serrano-Serrano et al. [22], excluding outgroups. All the analyzed fragments were used in the 29 sequences of the Valencian group with very low genetic diversity estimates. There were no gaps and 4080 sites, of which only four were variable and none showed any pattern of change. This group presented nine haplotypes, $3.8 \times 10^{-4}$ of nucleotide diversity and a non-significant genetic differentiation estimate $K_{st}$ of 0.022. The group including the South American varieties comprised 67 sequences of concatenated ITS, *Atpb-rbcL* and *trnL-trnF* fragments, 1800 sites and 1413 sites excluding gaps, 44 of which were variable. The group showed 37 haplotypes, a nucleotide diversity of $3.89 \times 10^{-3}$ and a significant genetic differentiation estimate $K_{st}$ of 0.528 (Table 5).

**Table 5.** The polymorphism and divergence data of the two *P. lunatus* cultivars groups. The Valencian landraces included the most frequently used cultivars in Spain ('Peru', 'Pintat', 'Ull de Perdiu', 'Cella Negra'). The Valencian + South American group included the Valencian and South American accessions from Serrano-Serrano et al. [22]. N: number of individuals, n: number of sites, n': number of sites excluding sites with gaps/missing data, S: number of variable sites, h: number of haplotypes, $\Pi$ (s.d.): nucleotide diversity and standard deviation in brackets, $H_d$ (s.d.): haplotype diversity and standard deviation in brackets, $K_{st}$: genetic differentiation estimate and its *p*-value (n.s.: non-significant, ***: $p < 0.001$).

| | N | n | n' | S | H | $\Pi$ (s.d.) | $H_d$ (s.d.) | $K_{st}$ |
|---|---|---|---|---|---|---|---|---|
| Valencian cultivars | 29 | 4080 | 4080 | 4 | 9 | $3.8 \cdot \times 10^{-4}$ ($4 \times 10^{-5}$) | 0.862 (0.035) | 0.022 n.s. |
| Valencian + S. American Accessions | 67 | 1800 | 1413 | 44 | 37 | $3.9 \cdot \times 10^{-3}$ ($2.5 \times 10^{-4}$) | 0.954 ($2 \cdot 10^{-4}$) | 0.528 *** |

## 4. Discussion

A landrace differs from a variety that has been selectively modified to improve particular characteristics. These traditional landraces, cultivated continuously for years, are severely threatened by genetic extinction because they are replaced with modern varieties, selected mainly for their higher productivity instead of their resistance to climate change consequences [43].

Currently, the commercial white-seed bean ('Peru') is the cheapest and the most widely sold among lima beans in the Valencian Community, and probably the only one known to most people. Seeds of 'Pintat', and rarely of 'Ull de Perdiu' are sold only in a few local markets, while the cultivar 'Cella Negra' has practically disappeared. The EEA-Carcaixent conserves and multiplies a few accessions of the cultivar 'Cella Negra' for its preservation, from the few seeds that it has been able to find from some farmers who still cultivate it for their own use. We focused our research according to the assumption that the commercial predominance of the different cultivars is not a question based on consumer preferences, but on local farmers' low profitability.

Local landraces are associated with one specific geographical location and, therefore, present climatic adaptability. They are generally better adapted to abiotic stress than modern cultivars [44] and supporting the recovery of their cultivation can mean advantages to face the climate change threat, especially if consumer demand increases. Hence, this climatic adaptability reveals the need to conserve the landrace germplasm as a means to provide information about adaptations to drought and heat stress, and because it constitutes a tool to identify stress-tolerant alleles to improve productivity when faced with climate change [45,46].

Baudoin [47], after thoroughly reviewing the diversity of *Phaseolus lunatus*, already indicated this species as an underexploited crop with a very high cultivation potential given its ability to withstand several types of stress, including severe drought. Moreover, the necessity of carrying out preservation programs for germplasm banks of wild forms and landraces was highlighted.

Regarding seed morphology, clear variability that depends on cultivars is described in the literature. Additionally, variations in dimensions, test patterns and color in cultivars from different countries

are known as Potato with small rounded seeds, along with Sieva, with medium-sized reniform seeds, while Andean ones are known as Big Lima and have large, but flat, seeds [14].

In the four studied landraces, no major differences in seed dimensions appeared. Seeds have the morphological characteristics of most of the individuals cultivated for commercial purposes, mainly with big, attractive and nutritional seeds, which indicate their Andean origin.

Regarding seed color and according to bibliographic references, the most frequent seed coat color of the cultivated plants differs depending on the considered geographic area. White seeds are one of the most frequently found among Cuban cultivars [48], while a predominant bicolored pattern is observed in the cultivars grown in Peru [12]. The studied landraces exhibit different bicolored patterns, which also agrees with their geographical provenance.

The seeds of the studied landraces underwent fast germination with no primary dormancy trait, even though dormancy was detected in some colored lima beans [49]. The marked preference for high temperatures in this species stood out, as clearly evidenced by our results with an optimal germination response at 30 °C. Indeed, Polock and Toole [50] and Polock [51] indicated that temperatures below 25 °C in the imbibition phase can be harmful. Other authors have indicated a good response of lima beans at high temperatures (25 °C and 30 °C), even when they were exposed to different salt concentration levels [52].

When we compared germination behavior of the commercial cultivar and the landrace 'Pintat' for the different tested temperature regimes, a stronger competitiveness of the local cultivar was observed from 19 °C to the optimum temperature, close to 30 °C.

Drought is one of the most important problems in agriculture as it leads to reduced yields and loss of crops. Water availability is essential for plants, as they need a good water supply throughout their life cycle. Therefore, water deficit in plants affects all phases of their development, physiological processes, growth and production which, under extreme conditions, can lead to plants dying [43,53,54]. Like the exposed response to temperatures, our results also support the hypothesis of the higher tolerance of those landraces cultivated for years that better adapt to changes in environmental conditions deriving from the Mediterranean climate. In fact, the 'Pintat' landrace was the most tolerant to water stress, simulated by lowering the osmotic potential of PEG solutions. In fact, the thermal time and hydrotime parameters have proven to be good discrimination tools to identify drought- and high temperature-tolerant common bean cultivars [55].

Conversely, DNA barcoding has not found any differences between the local and commercial cultivars used in the Valencian Community, not even when using different sorts: chloroplastic, nuclear, codificant and non-codificant markers. Nevertheless, their origin can be clearly situated. Recent phylogenetic studies have used genome-wide SNP markers polymorphisms [56] to indicate that the wild lima bean is structured into three gene pools, as previously proposed by Serrano-Serrano et al. [22]: the Mesoamerican one (MI); the Mesoamerican two (MII); the Andean one (AI). Their geographic ranges do not generally overlap. In addition, Chacón-Sánchez and Martínez-Castillo [56] also suggest the existence of another Andean gene pool (AII) in central Colombia. Our phylogenetic analyses, based on the data of Serrano-Serrano et al. [22], placed the Valencian cultivars in AI in relation to the 'Big Lima' morphology. These cultivars were phylogenetically grouped with the Andean Cordillera accessions from Ecuador, this being the domestication area of the Andean gene pool located between Ecuador and northern Peru [22,57].

For the Mesoamerican landraces, recent evidence indicates a scenario of a single domestication event in the gene pool MI for all the Mesoamerican landraces, perhaps in central-western Mexico, and the subsequent admixture among landraces and wild populations within the distribution range of gene pool MII, which gave rise to the MII landraces [56]. Therefore, and according to our results, these previous studies have shown that domestication was accompanied by strong founder effects that decreased the genetic diversity of the landraces in the Andes and MI of Mesoamerica. Thus, low polymorphism and divergence statistics have been found in the cultivars used in the Valencian Community (Spain), even between traditional ('Pintat', 'Ull de Perdiu', 'Cella Negra') and commercial ones ('Peru').

However, they all came from the same original gene pool in which a split occurred, as the earliest, when Europeans arrived in America 500 years ago, which is a negligible time in evolutionary terms.

Although other genome-wide barcoding techniques can be used [56,58], the different responses of these genetically close landraces can be explained by epigenetic mechanisms or by a few genes that play a relevant role in crop stress responses [59]. Indeed, rather than DNA barcoding, the search for relevant genes and local landrace alleles related to water stress tolerance could lead to new research works to help preserve these cultivars, by identifying the particular genetic features and their purity. When considering crop tolerance to overcome climate change-related stresses, natural variance among different cultivars can act as genetic reservoir for adaptation capability [60]. This idea, combined with an interest in providing added value to local landraces to defend their use recovery and agro-biodiversity conservation, could supply key future tools that promote local activities to face climate change effects on crops in order to contribute to the auto-sustainability of agronomy activities.

**Supplementary Materials:** The following are available online at http://www.mdpi.com/2073-4395/10/6/758/s1.

**Author Contributions:** Conceptualization, E.E., J.R. and P.S.; Investigation, M.I.M.-N., E.E. and P.S.; Methodology, M.I.M.-N., E.E., J.P.-M. and P.S.; Project administration, E.E. and P.S.; Visualization, M.I.M.-N., E.E. and P.S.; Writing—Original draft, M.I.M.-N., E.E. and P.S.; Writing—Review & Editing, M.I.M.-N., E.E., J.P.-M., J.R. and P.S. All authors have read and agreed to the published version of the manuscript.

**Funding:** This research received no external funding.

**Acknowledgments:** The authors wish to thank the Estación Experimental Agraria de Carcaixent (Valencia, Spain) for its support, particularly Fernando Amorós Ortega for providing us with some of the plant materials (seeds and leaves) used in the experiments. The authors sincerely acknowledge the valuable comments, corrections and suggestions made by anonymous reviewers that have significantly improve the manuscript.

**Conflicts of Interest:** The authors declare no conflict of interest.

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
