# Peer review of "Resilience Capacity Assessment of the Traditional Lima Bean (Phaseolus lunatus L.) Landraces Facing Climate Change"

_agronomy, doi:10.3390/agronomy10060758_

Round 1

Reviewer 1 Report

It is a well written paper. The author showed the importance of conserving the wild landraces diversity. It does a good job analyzing each beans on their physiology and genetic composition.

Line 16: What is PEG? You need to introduce this abbreviation first.

Line 43: Instead of saying wealthy nations, you could say “developed countries”

Line 60: The word “powerful” is a bit subjective, what specific property do you mean?

Figure 1: Nice photo. Instead of using abcd, you could just put the name of the cultivar in the picture

Line 194: Avoid using the word “good”, it is a subjective adjective, using some words more scientific

Line 210: What do you mean a more marked slope? Do you mean more “steep”?

Line 212: What is MGT? You need to introduce all abbreviation when you first mentioned it.

Figure 4 is a bit too small, please enlarge the graph.

I am not too sure about what the unit bar means in concentration

Line 317. Typo “Iwith”

Line 326. Maybe mentioned the author of [47] and [48] instead of just citing it.

Author Response

Thank you for your valuable comments. We have taken into account all the recommendations and modifications proposed, as detailed below.

Point 1: Line 16: What is PEG? You need to introduce this abbreviation first.

Response 1: We have added polyethylene glycol (PEG)

Point 2: Line 43: Instead of saying wealthy nations, you could say “developed countries”

Response 2: Wealthy nations has been changed to developed countries

Point 3: Line 60: The word “powerful” is a bit subjective, what specific property do you mean?

Response 3: Powerful has been substituted for strong to clarify the meaning of this sentence

Point 4: Figure 1: Nice photo. Instead of using abcd, you could just put the name of the cultivar in the picture

Response 4: The letters abcd have been replaced with the name of each cultivar. The figure caption has also been changed accordingly.

Point 5: Line 194: Avoid using the word “good”, it is a subjective adjective, using some words more scientific

Response 5: The sentenceGood results were achieved for the germination percentages at almost all the tested temperatures” has been replaced with: High germination percentages were achieved at almost all the tested temperatures.

Point 6: Line 210: What do you mean a more marked slope? Do you mean more “steep”?

Response 6: Yes, the words we have used are confusing. We have replaced them by” steeper slope”

Point 7: Line 212: What is MGT? You need to introduce all abbreviation when you first mentioned it.

Response 7: We have added “Germination Percentage and Mean Germination Time (MGT) were considered to compare seed responses” in lines 112-113.

Point 8: Figure 4 is a bit too small, please enlarge the graph.

Response 8: We have enlarged Figure 4. 

Point 9: I am not too sure about what the unit bar means in concentration

Response 9: This has been clearly an error, as bar is a unit of pressure used for osmotic potential, and thus, we have changed it in Figure 5, and also in the figure caption. As well, this mistake has been corrected in table 4 and in those phrases where this incorrectness has been detected.

Point 10: Line 317. Typo “Iwith”

Response 10: “Iwith” has been changed to “with”

Point 11: Line 326. Maybe mentioned the author of [47] and [48] instead of just citing it.

Response 11: We have replaced the sentence “Indeed [47] and [48] indicated” byIndeed, Polock and Toole [47] and Polock [48] indicated”

Reviewer 2 Report

Legumes are one of the most important crops that can contribute to mitigate climate change effects. All researchers focused on study climate change effects in crops, especially in legumes, are interesting for agriculture.

In order to improve the quality of this paper, I have include some questions in the pdf document to be considered by the authors. It is not clear for me how many accessions are analyzed in sum, including the phylogenetic analyses, and how many individuals per accession are analyzed. Material and methods section can be improved at this point.

In my opinion, the main limitation of this research is the low number of accessions considered (only 3 landraces and one commercial cultivar). In addition to that, germination time at different temperatures were estimated only in two cultivars and drought experiment neither is completed at all conditions due to lack of seeds. Taking this into consideration, seems difficult draw strong conclusions. It is important to work with local landraces, but maybe the conclusions can be softened accordingly to the low number of accessions studied.

Author Response

Thank you for your valuable comments. We have taken into account all the recommendations and modifications proposed, as detailed below.

Point 1: In order to improve the quality of this paper, I have included some questions in the pdf document to be considered by the authors. It is not clear for me how many accessions are analyzed in sum, including the phylogenetic analyses, and how many individuals per accession are analyzed. Material and methods section can be improved at this point. This information can be included in the material and methods section 2.3.1

Response 1: The Material and Methods section has been rewritten to clarify it and to add the information requested about the analized accessions.

In 2.3.1. We have included “Eight individuals of each cultivar were analyzed, except for ‘Cella Negra’, for which only five individuals were available at the time that the genetic assays were done. For ‘Ull de Perdiu’ and ‘Pintat’, we studied two different samples: one seed accession was obtained from the EEA-Carcaixent, while the other was bought from a traditional market. All the ‘Cella Negra’ seeds came from the EEA-Carcaixent and all the ‘Peru’ ones were obtained from a market as local farmers do not traditionally cultivate them. All the accessions were identified according to seeds’ distinctive morphological features”.

Moreover, in 2.3.3., we have specified the number of individuals used for that particular analysis: “Two individuals of each Valencian landrace and all the accessions provided by Serrano-Serrano et al. [22] in NCBI were analyzed.”

All the questions in pdf have been considered and corrected:

-The duplication in L9 has been eliminated

- L98: It should be great consider if all seeds derived from the same growing season. Germination capacity is strong influenced by the age of the seeds.

Yes, the seeds provided by the EEA-Carcaixent were collected during the previous season, nearly 1 year before germination tests started. We did not collect data of the seeds of commercial origin. We have indicated this in the text.

The differences observed in seed vigor between the different accessions could be due to either distinct maternal effects or maturity levels upon collection. We did not control this previous condition of the studied samples.

-We have corrected the letters in Table 3 and the figure caption to indicate homogeneous groups.

- L 260: How many individuals per cultivar were analyzed? This information can be included in the material and methods section 2.3.1

-The Material and Methods section has been modified and this question is already included.

Point 2: In my opinion, the main limitation of this research is the low number of accessions considered (only 3 landraces and one commercial cultivar). In addition to that, germination time at different temperatures were estimated only in two cultivars and drought experiment neither is completed at all conditions due to lack of seeds. Taking this into consideration, seems difficult draw strong conclusions. It is important to work with local landraces, but maybe the conclusions can be softened accordingly to the low number of accessions studied

Response 2: The objective of this study was, on the one hand, to value these traditional crops by demonstrating their greater capacity to face the environmental changes deriving from rising temperatures and water stress and, on the other hand, to transmit these results to the farmers on the territory, and even to consumers, to avoid the predominant use of commercial cultivars from other territories that are less adapted to the Mediterranean climate. Indeed, we considered in our research all the landraces traditionally used in eastern Spain. Currently, these cultivars are in regression and only used in small traditional farming areas, which makes it very difficult to obtain enough seeds to complete all the planned experiments.

Our main research achievement was to demonstrate the greater competitiveness of one of the traditional cultivars (Pintat) in terms of temperature and tolerance to water stress compared to the commercial one.

This study will, through the Carcaixent Agricultural Experiment Station, which is in direct contact with local farmers, convey the need to use local cultivars by contributing to promote agro-diversity

In any case, we have decided to eliminate the data of the cultivars with an insufficient number of seeds from both the test and figures, and to include them as Supplementary Material to simplify and avoid non conclusive data.

Reviewer 3 Report

Martinez-Nieto and collaborators present a well-conducted study comparing seed traits (size, color and germination success) in three landraces and one commercial variety of Lima bean, Phaseolus lunatus.  In addition, they performed genetic and phylogenetic analyses on these four varieties. They were specifically interested on the differential ability of these varieties to withstand abiotic stress mainly, increased temperatures and water stress, two of the most important factors that will likely increase with climate change.

Overall I think this is an interesting and well-conducted study.

I believe it would be a timely and important to both plant physiologists and agronomists. Nevertheless, I would like to highlight certain minor aspects in the manuscript that need consideration or further clarification.

  • The three landraces and the commercial variety: As I see it, the rational of the study is to identify the landrace that has those traits that would allow it a better survival under climatic change conditions. Because landraces are selected under local selective pressures it is then expected that one of the three landraces will be a more suitable choice than the commercial variety.
  • I would have liked to see more information on how and when these landraces were selected. Where the three selected in the same region? Under which conditions? For a specific reason? For example fast growth, taste, resistance to biotic stress?
  • The seeds were characterized for many traits. I was surprised that there is no mention on the chemistry. In this context, P. lunatus is particularly interesting because from the four species of Phaseolus that were domesticated, it is the only one that has cyanogenic glycosides (CNGs), and a higher content in the seeds compared to other plant structures (see references I added at the end of the review). These compounds are known for their defensive role against biotic stressors (insects and pathogens) and a recent paper showed that CNGs in the seeds may also play a role in seed germination and seedling growth.
  • Even if the chemistry was not part of the objectives of this study, it should at least be mentioned in the discussion as it is also relevant when comparing the adaptability of different varieties, for example trade-offs between germination and defense).
  • Along these lines, again even if it was not part of the objectives, I would have liked to see some mention how these landraces and commercial variety differ under biotic stress, pathogens or insects.

These last two comments are just suggestions, will not change the results or their interpretations but if included in the discussion would make the study more complete.

  • It was not very clear what is the relationship between the first part on characterization of seed traits and the genetic part. In L77 “…in an attempt to find the potential correlation of landraces genetic characteristics and germination responses”. The objective is interesting but I fail to see how this is again linked with the results and the discussion sections. What do the gentec and phylogenetic analyses tell you on seed germination? And how do you or can you use this information to select the “best” variety? There seems to be something missing there. As it is now the study comprises two disconnected parts that need to be integrated.

  • In the discussion L362: “Indeed the search for relevant genes…..could lead to new research works….” The statement makes sense and would be very interesting if this was accomplished but I do not see any evidence for this in your results. Please explain.

Minor comment:

L317.- at the beginning of the sentence there is an l that should not be there

Some references:

Cuny, etal. 2019. Role of cyanogenic glycosides in seeds of Lima bean (Phaseolus lunatus): Defense, plant nutrition or both?. PlantaDOI: 10.1007/s00425-019-03221-3

Gleadow R.M. & Woodrow I.E. (2002) Constraints on effectiveness of cyanogenic glycosides in herbivore defense. Journal of Chemical Ecology28, 1301–1313.

Author Response

Thank you for your valuable comments. We have taken into account all the recommendations and modifications proposed, as detailed below.

Point 1: I would have liked to see more information on how and when these landraces were selected. Where the three selected in the same region? Under which conditions? For a specific reason? For example, fast growth, taste, resistance to biotic stress?

Response 1: Yes, the three considered traditional cultivars are cultivated in the same region. We have included the following paragraph in the Introduction to clarify this aspect:

The use of this species in the eastern Iberian Peninsula in traditional cuisine is very ancient. Today we only have references of using these four cultivars in the past 100 years in this region.

Our main selection criterion for this work was to recover forgotten crops for the future. In fact, some of these studied cultivars, specifically 'Cella Negra', have practically disappeared today and it has been very difficult for us to find the seeds of this plant.

Point 2: The seeds were characterized for many traits. I was surprised that there is no mention on the chemistry. In this context, P. lunatus is particularly interesting because from the four species of Phaseolus that were domesticated, it is the only one that has cyanogenic glycosides (CNGs), and a higher content in the seeds compared to other plant structures (see references I added at the end of the review). These compounds are known for their defensive role against biotic stressors (insects and pathogens) and a recent paper showed that CNGs in the seeds may also play a role in seed germination and seedling growth.

Response 2: This could be an interesting point to consider in future research with this group, but unfortunately our background and laboratory aren’t appropriate to complete this kind of analysis. Future collaborations with other researchers experienced in this field could be very fruitful to develop this line. Thank you. We will seriously consider this suggestion.

Point 3: Even if the chemistry was not part of the objectives of this study, it should at least be mentioned in the discussion as it is also relevant when comparing the adaptability of different varieties, for example trade-offs between germination and defense).

Response 3: We completely agree with this comment and have incorporated additional texts in the Introduction accordingly. We have contemplated mentioning it in the discussion but finally we have considered it could be too extensive.

Point 4: Along these lines, again even if it was not part of the objectives, I would have liked to see some mention how these landraces and commercial variety differ under biotic stress, pathogens or insects.

These last two comments are just suggestions, will not change the results or their interpretations but if included in the discussion would make the study more complete.

Response 4: As in this study phase we focused only on seed response in the laboratory, we have no data on biotic stresses. We are aware of the importance of this aspect and will consider it for future field studies.

Point 5: It was not very clear what is the relationship between the first part on characterization of seed traits and the genetic part. In L77 “…in an attempt to find the potential correlation of landraces genetic characteristics and germination responses”. The objective is interesting, but I fail to see how this is again linked with the results and the discussion sections. What do the gentec and phylogenetic analyses tell you on seed germination? And how do you or can you use this information to select the “best” variety? There seems to be something missing there. As it is now the study comprises two disconnected parts that need to be integrated.

Response 5: Our aim in the molecular assays was to genetically characterize traditional Valencian cultivars by including their origin to delimitate landraces and to enable their purity to be tested in the future. At the same time, we have attempted to find a potential correlation between genetic characterization and germination response.

Unfortunately, as the phylogenetic analyses showed, their proximity complicated this task, even when studying different sorts of molecular markers. Nevertheless, some differences may appear as different responses were found in our analyses, apart from organoleptic features. For this reason, and in order to continue working on this objective, we think that it is important to publish these analyses.

However, a paragraph has been added to the Introduction and some nuances to the Discussion to connect the different work sections and to make it easier to understand.

We have attempted to find differences in genetic diversity that support the differences in germination behavior. We initially hoped to find higher diversity in those traditional cultivars to support greater tolerance to germination responses.

Point 6: In the discussion L362: “Indeed the search for relevant genes…could lead to new research works….” The statement makes sense and would be very interesting if this was accomplished but I do not see any evidence for this in your results. Please explain.

Response 6: As explained above, these cultivars share a very close phylogenetic relation. Moreover, it is important to consider that domestication was accompanied by strong founder effects that decreased genetic diversity (Chacón-Sánchez & Martínez-Castillo, 2017). For these reasons, the genetic characterization using ‘blind’ markers appeared to be a difficult mission, even when they have been useful for identifying wild intraspecific lineages (Serrano-Serrano et al, 2010). However, different responses have been found in our study and it is logical to think that certain local adaptations related to some genes/alleles could be found. Thus, we hypothesized that looking for specific genes and alleles which lead to these differences could shed light on the objective of the molecular characterization of landraces and to find a correlation between genetic characterization and germination responses. In order to clarify this argument in the Discussion, this part has been reviewed.

Minor comment:

L317.- at the beginning of the sentence there is an l that should not be there

Response: It has been removed.

Some references:

Cuny, et al. 2019. Role of cyanogenic glycosides in seeds of Lima bean (Phaseolus lunatus): Defense, plant nutrition or both?. PlantaDOI: 10.1007/s00425-019-03221-3.

Gleadow R.M. & Woodrow I.E. (2002) Constraints on effectiveness of cyanogenic glycosides in herbivore defense. Journal of Chemical Ecology28, 1301–1313.

The references have been considered and added.

Reviewer 4 Report

The manuscript entitled “Resilience capacity assessment of the traditional lima bean (Phaseolus lunatus L.) landraces facing climate change” 
investigated seed traits and germination rates in relation to temperature and water stress. It also analyzed molecular markers to calculate polymorphism and divergence. While this manuscript brings up important points in regard to the promotion of local landraces for improving biodiversity in agriculture, it is not publishable the way it currently stands.

The main issue is the lack of replicates for half of the cultivars being investigated. While it is unfortunate that more seeds for the missing cultivars could not be sourced, the data presented feels incomplete. It may be better to remove the two under-studied cultivars from the manuscript (or at the very least from the figures) and only compare the commercial cultivar against the ‘Pintat’ cultivar. Unfortunately, with so many missing replicates for the other two cultivars, the conclusions drawn from the germination data presented do not feel particularly supported.

The whole manuscript must also be reread for grammatical errors.

Some general comments are below:

MATERIALS AND METHODS

The degree symbol (°) here and throughout the entire manuscript looks like it may be underlined. Needs to be corrected

Line 95: Not sure what is meant by “previous tests”

Section 2.3: If the seeds of ‘Ull de Perdiu’ and ‘Pintat’ were sourced from different locations, how are you able to confirm they are indeed the same cultivar? How many plants did you extract DNA from for each cultivar? Was the DNA from 1 plant or did you pool DNA from multiple replicates?

RESULTS

It is stated that seed colour is an important consumer trait. Is there any more information on what consumers prefer and if consumers in different regions prefer different seed traits?

For the figures: Remove the Ull de Perdui and Cella Negra cultivars so the reader can easily compare the germination rates between the cultivars that were tested at all points.

Line 212- This is the first time that mean germination time as been abbreviated to MGT, need to clarify this in the text.

Figure 3: As a linear regression cannot be done for cultivars remove Ull de Perdui and Cella Negra, they should be removed from the figures.

Figure 6: Would benefit from a legend instead of just describing things in the text. What is the purple triangle depicting? Where do the cultivars described in this study fall?

DISCUSSION

Overall, the conclusions drawn from the data feels disjointed. There does not seem to be much connection between the seed features/germination assays and the genetic data presented, making the story being told in this manuscript difficult to follow.

Author Response

Thank you for your valuable comments. We have taken into account all the recommendations and modifications proposed, as detailed below.

Point 1: The main issue is the lack of replicates for half of the cultivars being investigated. While it is unfortunate that more seeds for the missing cultivars could not be sourced, the data presented feels incomplete. It may be better to remove the two under-studied cultivars from the manuscript (or at the very least from the figures) and only compare the commercial cultivar against the ‘Pintat’ cultivar. Unfortunately, with so many missing replicates for the other two cultivars, the conclusions drawn from the germination data presented do not feel particularly supported.

Reponse 1: Certainly, lack of seeds was the major problem with conducting the research. Initially, we decided to include the data of all the cultivars to highlight the different landraces that are currently in regression and constitute an important value to preserve the diversity of the traditional crops in this territory. In any case, after considering your comments and advice, we have decided to eliminate all the data corresponding to the cultivars with an insufficient number of seeds from the figures and text, and to include them as Supplementary Material as the final germination conclusions refer to the comparison of the ‘Pintat’ cultivar and the commercial ‘Peru’ one.

Point 2: The whole manuscript must also be reread for grammatical errors.

Response 2: The manuscript has been sent a second time to an English native speaker to correct possible grammar mistakes.

MATERIALS AND METHODS

Point 3: The degree symbol (°) here and throughout the entire manuscript looks like it may be underlined. Needs to be corrected

Response 3: We have corrected this throughout the manuscript.

Point 4: Line 95: Not sure what is meant by “previous tests”

Response 4: The first three paragraphs of this section have been restructured in an attempt to improve the understanding. This phrase has been removed to avoid possible confusion.

Point 5: Section 2.3: If the seeds of ‘Ull de Perdiu’ and ‘Pintat’ were sourced from different locations, how are you able to confirm they are indeed the same cultivar? How many plants did you extract DNA from for each cultivar? Was the DNA from 1 plant or did you pool DNA from multiple replicates?

Response 5: These cultivars are differentiated mainly by the morphological characteristics of seeds, and all the analyzed fragments were identical in most individuals. However, we have slightly modified the wording to clarify the text. Eight individuals were extracted for each cultivar, except for ‘Cella Negra’, as it has inserted in the text. Genetic analyses were done of individuals.

RESULTS

Point 6: It is stated that seed colour is an important consumer trait. Is there any more information on what consumers prefer and if consumers in different regions prefer different seed traits?

Response 6: We have included the following excerpts in the Discussion: Currently, the imported cultivar, with totally white seeds, is cheaper, is the most widely sold one, and probably the only one known to most people”. In a few local markets, it is possible to find seeds of 'Pintat', and rarely of 'Ull de Perdiu'. The cultivar 'Cella Negra' has practically disappeared from the market. The Carcaixent Experimental Agricultural Station conserves and multiplies a few accessions of the cultivar 'Cella Negra' for its conservation from a few seeds that it has been able to find from some farmers who still cultivate it for their own use. We focus our research according to the assumption that the commercial predominance of the different cultivars is not a question based on consumer preferences, but on local farmers’ low profitability. The advantages that local varieties can bring to face the climate change threat can support the recovery of their cultivation, especially if consumer demand increases.

Point 7: For the figures: Remove the Ull de Perdiu and Cella Negra cultivars so the reader can easily compare the germination rates between the cultivars that were tested at all points.

Response 7: The figures and figure captions about germination responses have been modified according to the comments.

Point 8: Line 212- This is the first time that mean germination time as been abbreviated to MGT, need to clarify this in the text.

Response 8: The following sentence has been added to Materials and Methods: “Germination Percentage and Mean Germination Time (MGT) were considered to compare seed responses”.

Point 9: Figure 3: As a linear regression cannot be done for cultivars remove Ull de Perdui and Cella Negra, they should be removed from the figures.

Response 9: The figures about germination responses, and also the figure captions, have been modified according to the comments.

Point 10: Figure 6: Would benefit from a legend instead of just describing things in the text. What is the purple triangle depicting? Where do the cultivars described in this study fall?

Response 10: A legend has been inserted into the figure that describes the meaning of the colors and triangles.

The following sentence has been introduced into the text to explain the meaning of purple triangles and to clarify where the studied cultivars fall: “the purple triangle represents the collapsed clades of Valencian landraces (local and ‘Peru’) as they were almost genetically identical”.

Point 11: Overall, the conclusions drawn from the data feels disjointed. There does not seem to be much connection between the seed features/germination assays and the genetic data presented, making the story being told in this manuscript difficult to follow.

Response 11. We have answered this same question to Reviewer 3 in point 5 by explaining the aim of the molecular assays and changes that we have made to improve this issue:

“Our aim in the molecular assays was to genetically characterize traditional Valencian cultivars, including their origin, to delimitate landraces and to enable the possibility of testing their purity in the future. At the same time, we attempted to find a potential correlation between genetic characterization and germination response.

Unfortunately, as the phylogenetic analyses showed, their proximity complicated this task, even when different sorts of molecular markers were studied. Nevertheless, some differences may exist as different responses were found in our analyses, apart from organoleptic features. For this reason, and in order to continue working on that objective, we think that is important to publish these analyses.

However, a paragraph has been added in the Introduction and some nuances in the Discussion to connect the different sections of the work and to make it easier to understand.”

DISCUSSION

Point 12: Overall, the conclusions drawn from the data feels disjointed. There does not seem to be much connection between the seed features/germination assays and the genetic data presented, making the story being told in this manuscript difficult to follow.

Response 12: As mentioned above, the Introduction and Discussion have been modified to make it easy to follow the argument.

Round 2

Reviewer 2 Report

No comments

Author Response

We acknowledge your valuable comments, which have contributed to improving the manuscript.

Reviewer 4 Report

The second version of the manuscript is much improved. My main concern is the writing of this version. While some parts read nicely, there are a few points (which I mainly noticed in the newly added passages) that contain significant grammatical issues which must be fixed before the manuscript can be published. I have outlines some points below but I think it needs to be proofread by a native English speaker before it can be published.

Some points:

Line 83: Sentence reading “Our main selection criterion for this work was to recover forgotten crops for the future. In fact, some of these studied cultivars, specifically ‘Cella Negra’, have practically disappeared today and has been very difficult for us to find seeds of this plant.” Needs rewording as it currently grammatically incorrect. I also do not think ‘selection criterion’ is the best word for here. ‘Aim’ may fit better.

Line 206: ‘While the other were bought’ should be ‘while the other was bought’

Section 3.2.2. As two of the cultivars have been removed, I would suggest modifying the language in this section as it previously had been written about multiple cultivars. For example, on line 451 ‘In all cases’ could be modified to ‘In both cases’. Additionally, ‘the best tolerance’ (line 462) suggests more than 2 cultivars, I would modify this to something like “ ‘Pintat’ demonstrated better tolerance to water stress…”

Line 523: ‘no germinating’ should be ‘non-germinating’

Figure 6: ‘Purple triangle’ has been written twice in the legend to refer to two different things

Paragraph on lines 673-680: First sentence needs to be reread for grammatical errors and sentence structure.

Line 678: ‘We focused our research according to assumption’ should be ‘we focused our research according to the assumption'

Author Response

We appreciate the time spent providing your valuable comments and suggestions, which have contributed to improving the manuscript. We have incorporated these changes to reflect the suggestions that you supplied.

Point 1. Line 83: Sentence reading “Our main selection criterion for this work was to recover forgotten crops for the future. In fact, some of these studied cultivars, specifically ‘Cella Negra’, have practically disappeared today and has been very difficult for us to find seeds of this plant.” Needs rewording as it currently grammatically incorrect. I also do not think ‘selection criterion’ is the best word for here. ‘Aim’ may fit better.

Response 1. The sentence has been revised and replaced by:

Our main aim in this work was to recover forgotten crops for the future. In fact, some of the studied cultivars, in particular ‘Cella Negra’, have practically disappeared today and it has been very difficult to find seeds of this plant.

Point 2. Line 206: ‘While the other were bought’ should be ‘while the other was bought’

Response 2. The word were has been replaced by was.

Point 3. Section 3.2.2. As two of the cultivars have been removed, I would suggest modifying the language in this section as it previously had been written about multiple cultivars. For example, on line 451 ‘In all cases’ could be modified to ‘In both cases’. Additionally, ‘the best tolerance’ (line 462) suggests more than 2 cultivars, I would modify this to something like “ ‘Pintat’ demonstrated better tolerance to water stress…”

Response 3. The sentence has been replaced with:

The cultivar ‘Pintat’ demonstrated better tolerance to water stress, which obtained values above 50% for the germination percentage for all the tested osmotic potentials up to -4 bar.

Point 4. Line 523: ‘no germinating’ should be ‘non-germinating’

Response 4. It has been corrected

Point 5. Figure 6: ‘Purple triangle’ has been written twice in the legend to refer to two different things

Response 5. Purple triangle has not been written twice, but "purple branches" and "purple triangle" refer to two related things. All purple branches refer to Andean I cluster where the Valencian landraces are included. For that reason, when collapsing the small group of Valencian samples, the triangle is also purple. However, the legend has been rewritten to clarify it:

“Colors correspond to the gene pools for wild P. lunatus: black branches belong to outgroups, purple branches to AI (Andean I) and the purple triangle inside represents the collapsed clade of Valencian landraces (local and ‘Peru’) as they were almost genetically identical, the orange triangle…”

Point 6. Paragraph on lines 673-680: First sentence needs to be reread for grammatical errors and sentence structure.

Response 6. The sentence has been modified according to the comments:

Currently, the commercial white-seed bean ('Peru') is the cheapest and the most widely sold among lima bean legumes in the Valencian Community, and probably the only one known to most people.

Point 7. Line 678: ‘We focused our research according to assumption’ should be ‘we focused our research according to the assumption'

Response 7. The sentence has been modified.
